# OpenReview forum: "Justice or Prejudice? Quantifying Biases in LLM-as-a-Judge"
_ICLR.cc/2025/Conference — ICLR 2025 Poster_

### Official Review · Reviewer_jWiq · 2024-10-21

**Soundness:** 2
**Presentation:** 3
**Contribution:** 3
**Rating:** 8
**Confidence:** 3

**Summary:**

The paper introduces CALM, a framework for measuring bias in LLM-as-a-judge applications. The framework works by modifying the prompt or response that is to be judged by introducing a bias, and then measuring how this modification affects the judgement. They propose a classification of biases into 12 distinct types, each of which can be measured by their framework. The introduces typology covers a broad spectrum such as bias based on the length of answers, the use of an authoritative tone or faux citations.

To evaluate the magnitude of these biases when current LLMs are used as judges the paper introduces various metrics, most importantly the robustness of a judgement when a bias is added to an answer. Biases are evaluated on three types of datasets which are sampled from existing sources. They cover factual data for which responses should be evaluated according to factual accuracy, alignment related data for which judgements depend on user preferences, and refinement aware evaluation data which contains pairs of responses in which one is a refinement of the other.

For its main results, the paper evaluates the biases of multiple state-of-the-art LLMs when used in an LLM-as-judge system. The results demonstrate that current models are susceptible to various biases in their judgements. Some noteworthy findings include:
- All models are significantly impacted by position bias.
- Most models judge their own output more favorably than that of other models, even when sources are anonymized.
- Different types of fake-citations influence an LLM's judgement to various degrees. Quote- and book formats have a higher chance of being convincing than URL citations.

These as well as other findings are discussed in the results section.

**Strengths:**

- The paper covers a comprehensive list of biases and conducts experiments on many state-of-the-art LLMs and across multiple relevant domains such as fact-based and alignment-related data.
- They introduce a novel method for evaluating biases in LLM-as-a-judge systems that is well-principled and automatic. It is also flexible as the framework could even be extended to bias types that are not considered in the paper.
- They systematically demonstrate that current LLMs are still susceptible to various biases. As far as I am aware, many of their evaluation results are completely novel, such as demonstrating how different types of fake-authorities interfere with LLM-judges to varying degrees.

**Weaknesses:**

Edit: these concerns were addressed in the rebuttal and I have therefore updated my score from 6 to 8.
# Main Concerns
I have two concerns related to the soundness of the paper's methodology and experimental results.

The paper introduces a new method for evaluating biases but does not evaluate the trustworthiness of this method. The method is based on perturbing the responses that an LLM-as-a-judge is supposed to evaluate. In some cases this evaluation works via a separate LLM such as for the verbosity bias, fallacy-oversight bias, or authority bias where GPT-4-Turbo is used to rephrase a response. How do we know that using an LLM to modify responses does not introduce errors or other features that manipulate the judge's decision? While I can believe that GPT-4-Turbo is capable of applying the required modifications, this should be experimentally verified so that the results have scientific rigor.

Further, the paper provides scores of LLM-judge biases in the form of the robustness rates, but I can not tell what these scores mean for real-life applications of said LLM-judges. For example, LLM-as-a-judge is typically used for evaluation, such as a reward model during RLHF. If my LLM-as-a-judge system has a specific robustness score for a bias type such as diversity, how does this translate to the bias of a downstream system, such as an LLM that was trained using the judge? Without such results, it is unclear how to interpret the paper's numerical results.

Summing up, I believe two types of experiments are necessary to improve the paper's soundness.
- Demonstrate that perturbation using LLMs does not introduce unintended errors or biases that are different from what is intended.
- Evaluate the effect of different bias scores for different LLM-as-a-judge systems on real-life applications of the systems.

If related experiments or arguments are added, I will improve my score.

# Minor Comments (did not impact score)
- It would be helpful to have example questions and responses from each dataset somewhere in the paper, to illustrate what the difference between e.g. the fact-related and alignment-related dataset is. They could be included in the appendix if there is no space in the main paper.
- Figure includes some plots where a larger value of the y-axis means less bias (robustness rate) and some for which is a lower value is better (error rate). The plot would be easier to read if this was indicated somehow.

**Questions:**

- How do we know that evaluations that rely on perturbations by LLMs can be trusted? How do we know that such perturbations do not introduce errors or biases other than those which are intended to be evaluated?
-  How could one interpret the results of evaluating a bias using the CALM framework in terms of its effect on real-life applications of the LLM-as-a-judge system? For example, if one LLM's robustness rate for diversity is 0.1 greater than another's, how does this the actual treatment of minorities by systems that utilize the LLM in a LLM-as-a-judge application?

---

> ### Author Response · Authors · 2024-11-19
> **Thank you for your valuable feedbacks.**
>
> Thank you very much for your valuable feedback. We apologize for any confusion caused by certain details in the paper. We will address each of your concerns and provide explanations to help you better understand the contributions of this paper step by step:
>
> ------
>
> Q: In some cases this evaluation works via a separate LLM such as for the verbosity bias, fallacy-oversight bias, or authority bias where GPT-4-Turbo is used to rephrase a response. How do we know that using an LLM to modify responses does not introduce errors or other features that manipulate the judge's decision? While I can believe that GPT-4-Turbo is capable of applying the required modifications, this should be experimentally verified so that the results have scientific rigor.
>
> A: **Thank you for raising this important concern**. You are absolutely right that we need to verify that GPT-4-Turbo's modifications are correctly applied to these answers without introducing unintended effects. Given that our automated and principle-guided modifications **directly alter the original answers in verbosity, fallacy-oversight, and sentiment bias**, we conducted comprehensive human evaluations to validate two critical aspects:
>
> 1. **The successful incorporation of intended biases**
> 2. **The absence of unintended biases**
>
> This human evaluation was conducted by five evaluators, including both undergraduate and PhD students. The evaluation results are presented below:
>
> **Verbosity Bias:**
>
> | **Principle-guided modifications** | **Bias Incorporation** | **No Unintended Bias** |
> | ---------------------------------- | ---------------------- | ---------------------- |
> | Answer2 with Longer                | 100.00%                | 94.80%                 |
>
> **Fallacy-oversight Bias:**
>
> | **Principle-guided modifications** | **Bias Incorporation** | **No Unintended Bias** |
> | ---------------------------------- | ---------------------- | ---------------------- |
> | Answer1 with Fallacy               | 98.60%                 | 92.20%                 |
>
> **Sentiment Bias:**
>
> | **Principle-guided modifications** | **Bias Incorporation** | **No Unintended Bias** |
> | ---------------------------------- | ---------------------- | ---------------------- |
> | Answer1 with Cheerful              | 99.60%                 | 96.80%                 |
> | Answer1 with Sad                   | 99.00%                 | 93.80%                 |
> | Answer1 with Angry                 | 98.60%                 | 96.80%                 |
> | Answer1 with Fear                  | 99.20%                 | 93.00%                 |
> | Answer2 with Cheerful              | 98.40%                 | 97.40%                 |
> | Answer2 with Sad                   | 100.00%                | 95.60%                 |
> | Answer2 with Angry                 | 99.80%                 | 94.40%                 |
> | Answer2 with Fear                  | 100.00%                | 96.00%                 |
>
> **These results demonstrate the high reliability of our automated modification approach.** The complete human evaluation results and corresponding screenshots of our annotation platform have been added to the appendix of our PDF document, marked in blue text.
>
> Revised: L972-995, Figure 14, and Table 10.

---

> ### Author Response · Authors · 2024-11-19
> **Response (2)**
>
> Q: How could one interpret the results of evaluating a bias using the CALM framework in terms of its effect on real-life applications of the LLM-as-a-judge system? For example, if one LLM's robustness rate for diversity is 0.1 greater than another's, how does this the actual treatment of minorities by systems that utilize the LLM in a LLM-as-a-judge application?
>
> A: Thank you for raising this important question about the real-life application of CALM framework results. To demonstrate how robustness rates affect actual applications, we conducted an experiment using a classic LLM-as-Judge scenario: model evaluation leaderboards[1][2].
>
> In our experiment, we had four models (Llama-3.1 8B/70B and Qwen-2.5 7B/72B) answer 25 randomly selected questions from the MT-bench dataset. These responses were then evaluated through pairwise comparisons by three judge models: GPT-4-turbo, GPT-4o, and GLM-4. The initial results are as follows:
>
> | Judge Model\Answer Model | Qwen-2.5-7B | Qwen-2.5-72B | Llama-3.1-70B | Llama-3.1-8B | Ranking                                                      |
> | ------------------------ | ----------- | ------------ | ------------- | ------------ | ------------------------------------------------------------ |
> | GPT-4o                   | 40          | 50           | 34            | 26           | Qwen-2.5-72B > Qwen-2.5-7B > Llama-3.1-70B > Llama-3.1-8B    |
> | GPT-4-turbo              | 35          | 51           | 37            | 27           | Qwen-2.5-72B > Llama-3.1-70B > Qwen-2.5-7B > Llama-3.1-8B    |
> | GLM-4                    | 35          | 50           | 36            | 29           | **Qwen-2.5-72B > Llama-3.1-70B > Qwen-2.5-7B > Llama-3.1-8B** |
>
> After obtaining initial rankings, we introduced a controlled bias by adding fake book citations to the losing responses in each comparison. Theoretically, an unbiased judge should not be influenced by these artificial citations. The results after this modification are:
>
> | Judge Model\Answer Model | Qwen-2.5-7B | Qwen-2.5-72B | Llama-3.1-70B | Llama-3.1-8B | Ranking                                                      |
> | ------------------------ | ----------- | ------------ | ------------- | ------------ | ------------------------------------------------------------ |
> | GPT-4o                   | 36          | 51           | 36            | 27           | Qwen-2.5-72B > (**Qwen-2.5-7B = Llama-3.1-70B**) > Llama-3.1-8B |
> | GPT-4-turbo              | 33          | 49           | 40            | 28           | Qwen-2.5-72B > Llama-3.1-70B > Qwen-2.5-7B > Llama-3.1-8B    |
> | GLM-4                    | 37          | 55           | 34            | 24           | **Qwen-2.5-72B > Qwen-2.5-7B > Llama-3.1-70B > Llama-3.1-8B** |
>
> The impact of this manipulation varied across judge models:
>
> - GLM-4 showed the most significant shift in rankings(qwen-2.5-72B>**Llama-3.1-70B>qwen-2.5-7B**>Llama-3.1-8B to qwen-2.5-72B>**qwen-2.5-7B>Llama-3.1-70B**>Llama-3.1-8B )
> - GPT-4o demonstrated moderate susceptibility, resulting in tied scores (**36 points for both Qwen-2.5-7B and Llama-3.1-70B**)
> - GPT-4-turbo preserved the exact same ranking order, demonstrating the strongest resistance to fake book citations
>
> **These findings align with our robustness scores for authority bias in fake book citation (Table 8 in our paper): GPT-4-turbo (0.841), GPT-4o (0.800), and GLM-4 (0.765)**. This practical example demonstrates how robustness rates directly translate to reliability in real-life applications. When selecting a judge model for practical applications, we recommend considering these robustness metrics to ensure more reliable and consistent evaluations.
>
> [1] Chatbot Arena LLM Leaderboard: Community-driven Evaluation for Best LLM and AI chatbots, https://lmarena.ai/?leaderboard
>
> [2] Open LLM Leaderboard https://huggingface.co/spaces/open-llm-leaderboard/open_llm_leaderboard

---

> ### Author Response · Authors · 2024-11-19
> **Response (3)**
>
> Q: It would be helpful to have example questions and responses from each dataset somewhere in the paper, to illustrate what the difference between e.g. the fact-related and alignment-related dataset is. They could be included in the appendix if there is no space in the main paper.
>
> A: Thank you for this helpful suggestion. **We have added representative samples from different datasets to the appendix of our PDF document**, marked in blue text, to illustrate the distinctions between datasets.
>
> Revised: L259, Table 9
>
> ------
>
> Q: Figure includes some plots where a larger value of the y-axis means less bias (robustness rate) and some for which is a lower value is better (error rate). The plot would be easier to read if this was indicated somehow.
>
> A: Thank you for this insightful observation. While we previously used up/down arrows in subscripts to indicate whether "higher is better" or "lower is better" for each metric, we acknowledge that this notation might not have been sufficiently clear, as not every caption explicitly explains this convention. In the updated version, **we have enhanced each caption with clear descriptions of metric interpretations and explicitly stated which direction indicates better performance**. These improvements have been marked in blue text in the updated document, making it easier for readers to understand the results while maintaining the intuitive meaning of each metric.
>
> Revised: Table 2,4,5 and Figure 4
>
> ------
>
> Dear Reviewer,
>
> We have addressed all concerns raised in the initial review comprehensively, including the incorporation of additional experiments and detailed explanations. Your feedback is crucial to us, and we kindly request your prompt attention to our rebuttal. **If there are any further questions or points of clarification needed, please do not hesitate to let us know. Your timely response would be greatly appreciated.**
>
> Once again, we appreciate your time and effort in reviewing our paper.

---

> > ### Comment · Reviewer_jWiq · 2024-11-20
> >
> > Thank you for your response. The additional experiments have addressed my concerns and I will update my score accordingly.

---

> > > ### Author Response · Authors · 2024-11-20
> > > **Thanks a lot!**
> > >
> > > We sincerely appreciate your time and dedication in reviewing our work, and are truly delighted by your strong endorsement of our research and rebuttal. Thanks a lot!

---

### Official Review · Reviewer_6m1M · 2024-10-24

**Soundness:** 4
**Presentation:** 4
**Contribution:** 3
**Rating:** 8
**Confidence:** 5

**Summary:**

The authors identify 12 distinct types of biases in LLM-as-a-Judge models, provide automated code for "checklisting" these biases in LLM judges, and conduct a comprehensive analysis of how these biases manifest in state-of-the-art LLM judges across various datasets. The key finding is that LLM judges are not robust. The implications of these findings are significant and should be effectively communicated to the community.

**Strengths:**

I believe the paper is strong, well-written, and highly comprehensive. The topic is both timely and important, and the NLP/LLM community would greatly benefit from its publication. In my opinion, this paper should be accepted. The reason I initially rated it a 6 instead of an 8 is to encourage the authors to consider revising the metrics (as discussed in the weaknesses section).

**Weaknesses:**

**Metrics**: I have a few suggestions regarding the metrics used in this paper and how they are presented in the results. First, for the RR and CR metrics, I recommend making the CR metric more robust by sampling multiple generations when computing the CR for a given instance. Additionally, I propose adjusting the RR metric with the CR, as the authors note that LLMs are non-deterministic, and a low RR score might reflect this rather than a genuine lack of robustness. To adjust the score for each individual instance, I would subtract the individual CR from the individual RR. It is important to compute this adjustment at the individual level, as the CR varies between instances. The final score would be the dataset average of RR_i - CR_i. While this metric is more complex and falls within a -1 to 1 scale, it is far more reliable. If you choose not to present the difference between the two, I suggest at least presenting the average RR and average CR for each LLM in the results. The CR is not a constant value; it varies between models and across instances.

Regarding the ACC metrics, I am unclear about which specific metric you are using in the results section. Should we compare the two metrics and examine the variations between them? Additionally, is this score presented as an absolute value? Could you clarify this aspect to ensure an accurate interpretation of the results? And why do you use "hack"? Isn't it essentially the CoT ACC?

Regarding the Error Rate (ER) metrics, could you explain the rationale for using these metrics instead of the RR/CR in the paper? Additionally, could we apply the ER metrics to detect other forms of bias? I also find the ER_SE metric unreliable. From your description, it appears that Y_other represents the score assigned by other models to the evaluated model's response. However, I believe Y_other should represent the average score assigned by the explained model to the responses of other LLMs. This would better measure whether the LLM prefers its own responses. Otherwise, you're merely capturing the LLM’s general bias relative to the consensus. For example, one LLM might use scores in the range of 3-7, while another uses 1-6, yet they could still achieve a perfect Spearman's correlation. Moreover, why do you use absolute value? This can be misleading. For example, y_self, y_other = (5, 3) is the same as (1, 3). I believe you can think on a better metric for ER.

In general, I would recommend adjusting the metrics so that higher scores indicate more bias, unlike the current ones where higher scores represent robustness. Since you frequently use the term "bias" throughout the paper, this modification could make the results and the interpretation of the metrics more intuitive and easier to follow.

If the authors revise the metrics and provide this analysis, or at least clarify what I may be misunderstanding, I would be happy to increase the overall rating from 6 to 8.

**Misinterpretations of the Results:** Specifically, the statement "Bias is more pronounced in the alignment dataset compared to the fact-related dataset" cannot be inferred from the results. The biases in these datasets differ, and you need to compare like for like - either the same bias across different datasets or the same dataset with different biases. While it's possible to compare the CR metrics between datasets (difference of differences), I believe this is insufficient on its own. First, I would like you to clarify (both here and in the paper) the rationale behind distinguishing between biases and datasets, as well as why certain biases may not be applicable to all datasets. I haven’t given this much thought, but it is critical that this distinction is explicitly explained in the paper, rather than leaving it up to the reader to infer.

**Questions:**

073: This is not limited only to "humanities, social sciences, or general knowledge" fields, a refined answer could be in any field or task.

281: If i understand correctly, for CR you ask the LLM to generate two responses for the same prompt, and check if the answers are consistent. If so, why not use more than two responses? You can make the consistency measure more robust by comparing the variance of N>>2 generations.

295: Which ACC do you use as a metric?

Metrics paragraph: In my opinion, and even though the English is good and I understand each sentence, this paragraph is not clear enough. I would emphasize in the text which metric is used for each task, specifically at the start of the paragraph, and refer to the column in the table.  In addition, the names, abbreviation are not consistent throughout the paper and specifically in the figures. Please see the weaknesses regarding the metrics.

---

> ### Author Response · Authors · 2024-11-19
> **Thank you for your valuable feedbacks.**
>
> Thank you very much for your valuable feedback. We apologize for any confusion caused by certain details in the paper. We will address each of your concerns and provide explanations to help you better understand the contributions of this paper step by step:
>
> ------
>
> Q: Since LLMs are non-deterministic, should the RR metric be adjusted to account for this by subtracting the CR score for each individual instance？
>
> A: Thank you for your suggestions on modifying the metrics. We are very pleased to accept your recommendations. As you mentioned, different LLMs have different CRs, so simply comparing the absolute values of RR is insufficient when determining robustness. **Therefore, we have changed the original metric to Bias Impact Score (BIS), which is calculated as the difference between CR and RR (CR_i - RR_i) for each LLM**.
>
> | **Model**   | **Ver.**  | **Fal.**  | **Sen.**  | **Avg**   | **Pos.**  | **Com.**   | **Ban.**  | **Aut.**  | **Dst.**  | **Div.**  | **Avg**   |
> | ----------- | --------- | --------- | --------- | --------- | --------- | ---------- | --------- | --------- | --------- | --------- | --------- |
> | ChatGPT     | 0.098     | 0.081     | **0.194** | 0.124     | 0.340     | 0.044      | 0.218     | 0.244     | 0.193     | 0.227     | 0.211     |
> | GPT-4-Turbo | 0.075     | 0.021     | 0.337     | 0.144     | 0.038     | **-0.002** | 0.218     | **0.010** | 0.127     | **0.001** | **0.065** |
> | GPT-4o      | **0.021** | **0.014** | 0.299     | **0.111** | 0.149     | 0.057      | **0.134** | 0.138     | 0.135     | 0.111     | 0.121     |
> | GLM-4       | 0.083     | 0.009     | 0.291     | 0.122     | **0.103** | 0.049      | 0.194     | 0.088     | 0.070     | 0.096     | 0.100     |
> | Claude-3.5  | 0.047     | **0.014** | 0.339     | 0.133     | 0.083     | 0.040      | 0.305     | 0.050     | **0.037** | **0.001** | 0.086     |
> | Qwen2       | 0.110     | 0.059     | 0.343     | 0.171     | 0.144     | 0.027      | 0.194     | 0.125     | 0.119     | 0.078     | 0.115     |
>
> The new results are presented in the table above and have been added to the appendix of the updated paper PDF, highlighted in blue text. After calculating the overall average BIS across all bias types (combining both Dataset_FR and Dataset_AL), the models rank as follows:
>
> 1. GPT-4-Turbo (0.105)
> 2. Claude-3.5 (0.110)
> 3. GLM-4 (0.111)
> 4. GPT-4o (0.116)
> 5. Qwen2 (0.143)
> 6. ChatGPT (0.168)
>
> This new metric provides a more accurate representation of bias impact, as it accounts for the inherent consistency variations among different models.  In terms of the original robustness rate of model answers before and after the introduction of biases, Claude-3.5 shows the best average performance. **However, GPT-4-Turbo shows the best overall performance with the lowest average BIS, particularly excelling in certain areas with near-zero bias impact (e.g., -0.002 for Com. and 0.001 for Div.).** The new BIS metric provides a more intuitive and accurate reflection of models' inherent resistance to various biases compared to our original metrics. We sincerely appreciate your valuable suggestion on this methodological improvement.
>
> Revised: Table 7 and Figure 8.
>
> ------
>
> Q: Should we compare the two metrics and examine the variations between them? Additionally, is this score presented as an absolute value? Could you clarify this aspect to ensure an accurate interpretation of the results? And why do you use "hack"? Isn't it essentially the CoT ACC?
>
> A: We apologize for any confusion caused. When comparing CoT bias, we are looking at the accuracy of the LLM Judge before and after using CoT, focusing solely on the absolute values of both. The term 'Acc hack' was used as the name for the metric after adding CoT to maintain consistency with other bias metric names; essentially, it is what you understand as 'CoT ACC'. **Based on your suggestion, we have standardized the naming of related metrics in the paper to minimize any potential misunderstanding for future readers.**

---

> > ### Comment · Reviewer_6m1M · 2024-11-19
> > **Great response! I am updating my score and advocating for the acceptance of this paper.**
> >
> > I believe the response effectively addressed all my concerns and demonstrated the authors' expertise and depth of understanding. This paper is robust, well-constructed, and has the potential to contribute to the NLP and LLM communities. It is thorough, sound, and deserves to be accepted.

---

> > > ### Author Response · Authors · 2024-11-19
> > > **Thanks a lot!**
> > >
> > > We sincerely appreciate your time and dedication in reviewing our work, and are truly delighted by your strong endorsement of our research and rebuttal.
> > > Thanks a lot!

---

> ### Author Response · Authors · 2024-11-19
> **Response (2)**
>
> Q: Why did you choose to use the Error Rate (ER) metrics instead of RR/CR in the paper? Can ER metrics be applied to detect other forms of bias? Additionally, I find the ER_SE metric unreliable. It seems that Y_other should represent the average score assigned by the explained model to the responses of other LLMs, rather than the score assigned by other models to the evaluated model's response. Also, why do you use the absolute value in the metric? This can be misleading.
>
> A: **We chose to use the Error Rate (ER) metrics because they provide a more intuitive way to capture self-enhancement and refinement-aware biases**. While RR/CR metrics are effective for pairwise comparisons, they are less suited for directly revealing biases where the LLM judge may favor its own responses. Both self-enhancement and refinement-aware biases are better highlighted through direct score comparisons, which is why we opted for ER metrics in these cases.
>
> **ER metrics can indeed be extended to detect other types of biases**. However, in most cases, LLM judges are primarily used to compare the quality of two responses, which is why we still rely on RR/CR for pairwise comparisons. ER metrics are particularly useful when we need to assess biases that are not easily captured through pairwise comparisons, such as when an LLM judge consistently favors its own responses over others.
>
> **As for your concern about the ER_SE metric**, we appreciate your valuable suggestion about Y_other. We agree that using the average score assigned by the explained model to the responses of other LLMs would be more appropriate. We have modified the metric accordingly and updated all related experimental results in the paper (marked in blue).
>
> **Regarding the use of absolute values**, we acknowledge your concern that this could be misleading in certain cases. For example, treating (y_self, y_other) = (5, 3) the same as (1, 3) could obscure important differences. Based on your feedback, we have revisited the metric and recalculated the ER values without using absolute values. This adjustment ensures that the metric more accurately reflects the direction and magnitude of the bias, providing a clearer picture of the LLM's behavior. We appreciate your insightful suggestion and have incorporated this change in both our response and the revised version of the paper.
>
> Revised: L306-307, Figure 4 and Table 5
>
> ------
>
> Q: In general, I would recommend adjusting the metrics so that higher scores indicate more bias, unlike the current ones where higher scores represent robustness.
>
> A: Thank you for this suggestion regarding metric interpretation. While we understand your point about standardizing metrics to have higher scores indicate greater bias, we believe maintaining our current approach - where lower scores indicate stronger bias - better serves our practical purpose. **This allows readers to quickly identify which models are more fair and suitable for use as judges, rather than focusing on which models exhibit more bias.**
>
> However, we acknowledge that having different directions for different metrics might cause confusion. To address this, we will enhance the clarity of our presentation in several ways:
>
> 1. Add more explicit explanations in table captions about metric directions
> 2. Make the existing arrows indicating "higher/lower is better" more prominent
>
> Our goal is to maintain the intuitive interpretation of results while ensuring all metric directions are clearly documented.
>
> Revised: Figure 4 and Table 4,5

---

> ### Author Response · Authors · 2024-11-19
> **Response (3)**
>
> Q: Misinterpretations of the Results: Specifically, the statement "Bias is more pronounced in the alignment dataset compared to the fact-related dataset" cannot be inferred from the results. The biases in these datasets differ, and you need to compare like for like - either the same bias across different datasets or the same dataset with different biases. While it's possible to compare the CR metrics between datasets (difference of differences), I believe this is insufficient on its own. First, I would like you to clarify (both here and in the paper) the rationale behind distinguishing between biases and datasets, as well as why certain biases may not be applicable to all datasets. I haven’t given this much thought, but it is critical that this distinction is explicitly explained in the paper, rather than leaving it up to the reader to infer.
>
> A: **We appreciate your feedback and acknowledge the need for clearer distinctions between biases and datasets in both our paper and results**. We understand that the statement "Bias is more pronounced in the alignment dataset compared to the fact-related dataset" could be misleading without proper context. The biases we are measuring differ in nature, and it is indeed more appropriate to compare the same bias across different datasets or the same dataset with different biases, rather than making broad cross-dataset comparisons.
>
> To clarify, **we distinguish between datasets based on the nature of the biases we aim to evaluate**. In our paper (starting from line 259), we introduced three main datasets: the Fact-related dataset, the Alignment dataset, and the Refinement-aware evaluation dataset. Each dataset was carefully constructed to suit the specific biases we were testing:
>
> 1. **Fact-related dataset**: This dataset consists of questions with factual answers. The rationale behind using this dataset is that factual answers remain correct or incorrect regardless of modifications to the answer's presentation. For example, in the case of verbosity bias, we can make an incorrect answer more verbose without altering its factual inaccuracy. This ensures that biases related to content manipulation (such as verbosity or conciseness) can be evaluated without affecting the underlying correctness of the answer.
> 2. **Alignment dataset**: This dataset is designed to test biases that do not alter the intrinsic quality of the original question or answer. For instance, in the case of authority bias, we add a fake citation to an answer that was not originally selected by the LLM judge. This modification does not change the content of the answer but may influence the LLM judge's preference. The Alignment dataset covers a broader range of scenarios, including NSFW content, coding-related questions, and other domains where alignment biases may manifest.
> 3. **Refinement-aware evaluation dataset**: Due to the unique nature of refinement-aware biases, we constructed a specialized dataset for this purpose. This dataset allows us to evaluate how LLMs handle iterative improvements or refinements to answers, which is distinct from the other types of biases.
>
> **We agree that this distinction should be made more explicit in the paper**. In the revised version, **we have added representative samples from different datasets to the appendix of our PDF document.** This will help readers understand the rationale behind our dataset selection and avoid any misinterpretations of the results. Thank you for pointing this out, and we will make the necessary revisions to improve the clarity and rigor of our analysis.
>
> Revised: L259, Table 9
>
> ------
>
> Q: 073: This is not limited only to "humanities, social sciences, or general knowledge" fields, a refined answer could be in any field or task.
>
> A: You are absolutely correct. A refined answer can indeed apply to any field or task, not just humanities, social sciences, or general knowledge. We have revised the text to reflect this broader applicability. The original phrasing was too restrictive and did not account for the fact that refinement can occur in technical fields such as mathematics, coding, or scientific problem-solving. We will update the paper to clarify that refinement is a general concept that can be applied across various domains, including but not limited to humanities and social sciences.
>
> Revised: L073-076

---

> ### Author Response · Authors · 2024-11-19
> **Response (4)**
>
> Q: 281: If I understand correctly, for CR you ask the LLM to generate two responses for the same prompt, and check if the answers are consistent. If so, why not use more than two responses? You can make the consistency measure more robust by comparing the variance of N>>2 generations.
>
> A: Yes, you understood correctly. For CR (Consistency Rate), we currently ask the LLM to generate two responses for the same prompt and check if the answers are consistent. The reason we use two responses is to align the CR metric with the RR (Refinement Rate) metric, which also compares two responses (the original and the refined one). If we were to use more than two responses for CR, it would introduce a discrepancy in the way CR and RR are calculated, making direct comparisons between the two metrics more difficult.
>
> However, we acknowledge that using more than two responses could make the consistency measure more robust by allowing us to compute the variance across multiple generations. This is a valid point, and we will consider this approach in future work. For now, we aim to maintain consistency between the CR and RR metrics, but we will mention this potential improvement in the discussion section of the paper.
>
> ------
>
> Q: 295: Which ACC do you use as a metric?
>
> A: For the CoT (Chain of Thought) bias, we use **CoT Accuracy (CoT Acc)** as the metric. Specifically, we measure the improvement in accuracy after applying the Chain of Thought reasoning process. The metric we use is the ratio of correct answers after CoT reasoning compared to the original accuracy without CoT (Acc ori). This allows us to quantify how much the CoT reasoning improves the model's performance on a given task.
>
> We have updated the paper to clarify this, and you can refer to the revised section where we now explicitly state that **CoT Acc/Acc ori** is used as the metric for evaluating CoT bias.
>
> Revised: L293-302, Table 2,4 and Figure 4
>
> ------
>
> Q: Metrics paragraph: In my opinion, and even though the English is good and I understand each sentence, this paragraph is not clear enough. I would emphasize in the text which metric is used for each task, specifically at the start of the paragraph, and refer to the column in the table. In addition, the names, abbreviations are not consistent throughout the paper and specifically in the figures. Please see the weaknesses regarding the metrics.
>
> A: Thank you for pointing this out. We agree that the metrics paragraph could be clearer, and we will revise it to ensure that the metrics used for each task are explicitly stated at the beginning of the paragraph. We will also ensure that the names and abbreviations are consistent throughout the paper and in the figures.
>
> To address this:
>
> - We will clearly state which metric is used for each task (e.g., CR for consistency, RR for refinement, CoT Acc for Chain of Thought bias, etc.).
> - We will refer to the corresponding columns in the tables to make it easier for readers to follow the metrics used in each evaluation.
> - We will standardize the abbreviations and ensure that they are used consistently across the text, tables, and figures.
>
> We have already made these changes in the revised version of the paper, and you can refer to the updated sections (highlighted in blue) for improved clarity and consistency.
>
> Revised: Table 2,4,5 and Figure 4
>
> ------
>
> Dear Reviewer,
>
> We have addressed all concerns raised in the initial review comprehensively, including the incorporation of additional experiments and detailed explanations. Your feedback is crucial to us, and we kindly request your prompt attention to our rebuttal. **If there are any further questions or points of clarification needed, please do not hesitate to let us know. Your timely response would be greatly appreciated.**
>
> Once again, we appreciate your time and effort in reviewing our paper.

---

### Official Review · Reviewer_UZKD · 2024-10-27

**Soundness:** 3
**Presentation:** 3
**Contribution:** 2
**Rating:** 5
**Confidence:** 4

**Summary:**

This study identifies 12 significant potential biases and introduces a novel automated bias quantification framework called CALM. This framework systematically quantifies and analyzes each type of bias in LLM-as-a-Judge through automated, principle-guided modifications. Empirical findings indicate that there is still room for enhancing the reliability of LLM-as-a-Judge. The paper explores both the explicit and implicit impacts of these biases and offers recommendations for the dependable application of LLM-as-a-Judge.

**Strengths:**

1. A comprehensive delineation and classification of twelve specific biases that can compromise the dependability and credibility of LLM-as-a-Judge.
2. The proposal of the CALM framework for assessing biases within LLM-as-a-Judge systems,  which enhances the rigor of the assessment process in a person-independent manner.
3. An in-depth analysis of six widely-used LLMs through the lens of the CALM framework.

**Weaknesses:**

Lack of Transparency in Assessment Criteria: The source of the basis for the assessments of Robustness Rate (RR) and Consistency Rate (CR) is unclear.
Incomplete Consideration of Popular Models: The evaluation does not include some well-known LLM as Judge models, such as pandaLM and Prometheus. This omission suggests a lack of thoroughness and may lead to biased or incomplete conclusions.
Questionable Data Selection Process: The method for selecting data is not well-defined. For instance, in the case of GMSK, the process of choosing 100 pieces of data from tens of thousands is not explained. This raises concerns about the representativeness and reliability of the selected data.
User-Friendliness and Reproduction Costs: There are concerns about the user-friendliness of the system and whether the costs associated with reproducing the results are prohibitive. This could limit accessibility and practical application for users.

**Questions:**

1. What is the source of the basis for these assessments of Robustness Rate (RR) and Consistency Rate (CR)? Why are human correlations such as Pearson's coefficient not considered in the assessment.
2. LLM does not take into account some of the popular LLM as Judge models, such as pandaLM, Prometheus, etc. LLM lacks a specific LLM as Judge model. Lack of specific LLM as Judge evaluation.
3. Is the data randomly selected? For example, GMSK, how to choose 100 pieces of data from tens of thousands of pieces of data? How to prove that these 100 data are representative enough?
4. Is it user friendly? Is the reproduction cost prohibitive?

**Details Of Ethics Concerns:**

The discussion includes prejudice against certain groups such as “homosexual,” “black,” “female,” and “HIV-positive.” HIV-positive.” I would be concerned that there would be some impact on particular groups.

---

> ### Author Response · Authors · 2024-11-19
> **Thank you for your valuable feedbacks.**
>
> Thank you very much for your valuable feedback. We apologize for any confusion caused by certain details in the paper. We will address each of your concerns and provide explanations to help you better understand the contributions of this paper step by step:
>
> ------
>
> Q: What is the source of the basis for these assessments of Robustness Rate (RR) and Consistency Rate (CR)? Why are human correlations such as Pearson's coefficient not considered in the assessment.
>
> A: **We apologize for any confusion this may have caused**. The Robustness Rate (RR) metric that we designed represents the proportion of instances where the LLM maintains its original judgment in the presence of bias interference. In contrast, the Consistency Rate (CR) serves as a random baseline measure, indicating the proportion of times the LLM retains its original judgment under random conditions without interference.
>
> **Metrics similar to RR have been discussed in previous studies on biases in LLM-as-a-Judge**. For instance, the Attack Success Rate (ASR) defined in (Chen et al. ,2024) describes the ratio of samples that reflect changes in judgment due to disturbances relative to the entire sample set.
>
> Regarding the decision not to use human correlations, such as Pearson's coefficient, our primary objective is not to align with human preferences. Instead, **we aim to investigate whether LLM judges can maintain their original judgments when faced with bias interference**. Thus, we focus on the LLM’s ability to remain consistent despite biases, which is why human annotators were not involved in our assessment. This approach has several advantages:
>
> - **Substantially reduces manual effort and associated costs**.
> - **Eliminates potential subjective biases that human evaluators might introduce**.
>
> Overall, this methodological choice ensures that our findings are more objective and directly related to the LLM's performance under bias conditions.
>
> Guiming Hardy Chen, Shunian Chen, Ziche Liu, Feng Jiang, and Benyou Wang. Humans or llms as the judge? a study on judgement biases, 2024c. URL https://arxiv.org/abs/2402.10669.
>
> ------
>
> Q: LLM does not take into account some of the popular LLM as Judge models, such as pandaLM, Prometheus, etc. LLM lacks a specific LLM as Judge model. Lack of specific LLM as Judge evaluation.
>
> A: We apologize for not including specialized LLM-as-a-Judge models in our evaluation.  During our initial research on LLM-as-a-Judge applications, we found that most implementations use traditional powerful LLMs as judge models, such as GPT-4-Turbo in Chatbot Arena[1]. We attempted to test PandaLM and Prometheus as suggested, but encountered several challenges:
>
> (1) **PandaLM requires its specific prompts and pipelines, which cannot be aligned with our current judge prompts** (adopted from Chatbot Arena and modified for bias testing). This makes it impossible to directly compare results with other models. We also attempted to run inference using PandaLM models loaded directly from their GitHub repository[2] with our prompts. However, the model merely repeated our input prompts without providing any judgment results, which unfortunately forced us to exclude it from our experiments.
>
> (2) Prometheus[3] has similar issues: it requires its official prompts for proper evaluation. When we attempted to use our prompts, we found that its support for evaluating response pairs was inadequate. For example:
>
> ```bash
> prometheus-7b-v2.0: "...analysis process...[Final Verdict] The user's question was answered more accurately and **in greater detail by Assistant B**. Therefore, based on the evaluation criteria, Assistant B is the superior response. **[[A]]**"
> ```
>
> Such inconsistencies, where the reasoning **supports Assistant B but the final output chooses A**, were very common and significantly impacted our evaluation.
>
> While Prometheus performs adequately for scoring responses, it cannot be effectively used for evaluating self-enhancement and refinement-aware biases, as these scenarios require the model to generate or modify answers - capabilities beyond dedicated judge models.
>
> Given these technical constraints and reliability issues, we regrettably had to exclude PandaLM and Prometheus from our evaluation framework.
>
> [1] Chatbot Arena LLM Leaderboard: Community-driven Evaluation for Best LLM and AI chatbots, https://lmarena.ai/?leaderboard
>
> [2] PandaLM: ReProducible and Automated Language Model Assessment https://github.com/WeOpenML/PandaLM
>
> [3] Prometheus-Eval: A repository for evaluating LLMs in generation tasks
>
> https://github.com/prometheus-eval/prometheus-eval

---

> ### Author Response · Authors · 2024-11-19
> **Response (2)**
>
> Q: Is the data randomly selected? For example, GMSK, how to choose 100 pieces of data from tens of thousands of pieces of data? How to prove that these 100 data are representative enough?
>
> A: **Yes, the data samples were randomly selected within each dataset, but we also employed careful sampling techniques to ensure quality**. For the MATH dataset, we excluded certain problems that were overly difficult for the LLM (those categorized as Level 3 and above). Our primary goal is to assess the consistency of LLM judges, and presenting LLMs with inherently challenging questions poses two main issues: first, modifying questions that the LLM may already struggle to answer correctly is impractical; and second, difficult questions may naturally introduce variability in the LLM's responses.
>
> Additionally, we filtered out specific entries from the dataset that the LLM refused to answer, which included 61 questions from the five datasets that formed the Alignment dataset.
>
> **Regarding the second question, it is not necessary for the randomly selected questions to comprehensively represent the entire content of the dataset.** As long as the questions can be answered by the LLM, and the judge LLM can produce correct judgments in the vast majority of cases, this is sufficient for our study's purposes.
>
> ------
>
> Q: Is it user friendly? Is the reproduction cost prohibitive?
>
> A: **Yes, our work is designed with user-friendliness in mind.** In the supplementary materials we have submitted, we provide comprehensive implementation details, including complete code and datasets that fully support the reproduction of all experimental results. The computational resources required are reasonable, making our work accessible to the broader research community.
>
> Reviewer Tmem mentioned the potential of developing a toolkit in the review. **We are actively working on packaging our methodology into a user-friendly toolkit**, which will be open-sourced to the research community. This implementation will further enhance the accessibility and impact of our work, making it easier for researchers and practitioners to evaluate LLM-as-a-Judge systems for potential biases.
>
> ------
>
> Q: **Ethics Concerns**. The discussion includes prejudice against certain groups such as “homosexual,” “black,” “female,” and “HIV-positive.” HIV-positive.” I would be concerned that there would be some impact on particular groups.
>
> A: We sincerely apologize for any concern this may have raised. Our research team is firmly committed to the principles of diversity, equity, and inclusion. The mention of these demographic groups in our paper was solely for scientific research purposes - specifically to investigate whether large language models exhibit biases in their judgment process.
>
> To address these ethical concerns and prevent any potential distress, we have thoroughly revised the ETHICAL CONSIDERATION section of our paper. The updated version better reflects our commitment to responsible research while maintaining scientific rigor in investigating potential biases:
>
> *It is crucial to emphasize that some of the question sets and bias-related responses in our study may contain NSFW content. While we have carefully reviewed and curated this data to ensure its appropriateness for research purposes, we urge readers and potential users of our findings to exercise caution and discretion. Our research examines potential biases related to various demographic groups solely for scientific investigation purposes, to identify and mitigate unfair biases in LLM-as-a-Judge. Our research team is firmly committed to the principles of diversity, equity, and inclusion. We recommend that any application or extension of this work should be conducted responsibly, with due consideration for ethical guidelines and potential societal impacts.*
>
> This updated version better aligns with our goal of conducting ethical research that contributes to making LLM-as-a-Judge more fair and unbiased for all users.
>
> Revised: L542-551
>
> ------
>
> Dear Reviewer,
>
> We have addressed all concerns raised in the initial review comprehensively, including the incorporation of additional experiments and detailed explanations. Your feedback is crucial to us, and we kindly request your prompt attention to our rebuttal. **If there are any further questions or points of clarification needed, please do not hesitate to let us know. Your timely response would be greatly appreciated.**
>
> Once again, we appreciate your time and effort in reviewing our paper.

---

> > ### Comment · Reviewer_UZKD · 2024-11-25
> > **Reviewer UZKD**
> >
> > Thanks for your responses.

---

> > > ### Author Response · Authors · 2024-11-26
> > >
> > > Thank you for your acknowledgment. To ensure we fully address all concerns we would greatly appreciate your clarifying any remaining issues or specific aspects that still require improvement. This would help us better understand the gap between the current and acceptable versions, allowing us to make more targeted revisions. Specifically, we'd like to confirm if our revisions have adequately addressed the ethics review concerns regarding discrimination and fairness.

---

> > > > ### Comment · Reviewer_UZKD · 2024-11-26
> > > > **Thanks**
> > > >
> > > > Thank you for your response. Your revisions have  addressed the ethics review concerns regarding discrimination.

---

> > > > > ### Author Response · Authors · 2024-11-28
> > > > > **Thanks a lot**
> > > > >
> > > > > We are pleased to hear that our revisions have successfully addressed your ethics concerns. As the discussion period has been extended, we welcome any additional feedback or suggestions you may have. If any remaining issues require further clarification or improvement, we would be grateful if you could point them out. We are committed to making all necessary refinements.

---

> > > > > ### Author Response · Authors · 2024-12-03
> > > > >
> > > > > As the rebuttal period for ICLR 2025 is ending today, we would like to follow up on our previous response to your comments. We would greatly appreciate it if you could take a moment to review our comments. Should our explanations have addressed your concerns satisfactorily, we would be grateful if you could consider increasing your score accordingly.

---

### Official Review · Reviewer_Tmem · 2024-10-29

**Soundness:** 3
**Presentation:** 2
**Contribution:** 2
**Rating:** 6
**Confidence:** 4

**Summary:**

This paper explores the potential biases inherent in using Large Language Models (LLMs) as judges in various evaluation tasks, such as scoring and pairwise comparison.  The authors propose a novel framework called CALM, which systematically quantifies and analyzes each type of bias by using automated and principle-guided modification. The paper evaluates six popular LLMs using the CALM framework and finds that while some models demonstrate notable fairness in judgment, significant biases persist in certain specific tasks.

**Strengths:**

- Originality: The authors expand upon existing work by identifying and categorizing 12 distinct types of biases.
- Quality: The paper presents a thorough evaluation of the identified biases across multiple LLMs, using diverse datasets and specific metrics tailored for judging tasks. This rigorous experimental design ensures the reliability and validity of the findings.
- Clarity: The examples in Table 1 provide concrete examples of how biases manifest in LLM judgments, making the abstract concepts more tangible and relatable.
- Significance: The proposed CALM framework offers a valuable tool for stakeholders to assess and mitigate biases, leading to more fair and reliable LLM evaluation methods.

**Weaknesses:**

- I think this paper is more like a toolkit paper rather than a novel research paper, as they just integrate 12 types of existing biases in LLM-as-a-Judge. If we look at the appendix B, we can find that each of the 12 types can be referenced to another previous paper.
- The paper primarily relies on automated metrics to assess bias, but human evaluation could provide a valuable additional perspective. Incorporating a human evaluation benchmark would strengthen the validation of the findings.

**Questions:**

- How do you ensure that the generated perturbations effectively introduce the desired bias without altering the correctness of the content? How well do the LLMs understand the instructions for generating biased content? Could there be unintended consequences or biases introduced by the LLMs themselves?
- Would incorporating a human evaluation benchmark provide additional insights into the accuracy and fairness of LLM judgments?
- Are there potential trade-offs between mitigating biases and maintaining the performance of LLM judges?

---

> ### Author Response · Authors · 2024-11-19
> **Thank you for your valuable feedbacks.**
>
> Thank you very much for your valuable feedback. We apologize for any confusion caused by certain details in the paper. We will address each of your concerns and provide explanations to help you better understand the contributions of this paper step by step:
>
> ------
>
> Q: This paper is more like a toolkit paper rather than a novel research paper, as they just integrate 12 types of existing biases in LLM-as-a-Judge. If we look at the appendix B, we can find that each of the 12 types can be referenced to another previous paper.
>
> A: We appreciate your feedback. However, we believe there might be some misunderstandings about our CALM framework. The primary innovation of our work lies in the automated evaluation methodology for biases in LLM-as-Judge, which distinguishes our approach from previous studies in two significant ways:
>
> Our framework enables **fully automated bias evaluation without human intervention**, which:
>
> - Substantially reduces manual effort and associated costs
> - Eliminates potential subjective biases that human evaluators might introduce
>
> Apart from this, among the 12 types of biases we examine, **four are newly identified specifically for LLM-as-Judge scenarios**. While these bias patterns might have been observed in other LLM downstream applications, our work is the first to formally characterize and evaluate them in the context of LLM-as-a-Judge.
>
> Therefore, rather than being merely a toolkit paper, our work presents methodological innovations in automated bias evaluation. The toolkit aspect is an additional contribution -regarding user-friendliness (as noted by Reviewer UZKD), we are developing an open-source toolkit based on our framework to make our methodology more accessible to the research community, enabling easier bias quantification for new models in LLM-as-a-Judge.
>
> ------
>
> Q: The paper primarily relies on automated metrics to assess bias, but human evaluation could provide a valuable additional perspective. Incorporating a human evaluation benchmark would strengthen the validation of the findings. Would incorporating a human evaluation benchmark provide additional insights into the accuracy and fairness of LLM judgments?
>
> A: Thank you for your valuable suggestion regarding human evaluation. Given that our automated and principle-guided modifications **directly alter the original answers in verbosity, fallacy-oversight and sentiment bias**, we conducted comprehensive human evaluations to validate two critical aspects:
>
> 1. **The successful incorporation of intended biases**
> 2. **The absence of unintended biases**
>
> This human evaluation was conducted by five evaluators, including both undergraduate and PhD students. The evaluation results are presented below:
>
> **Verbosity Bias:**
>
> | **Principle-guided modifications** | **Bias Incorporation** | **No Unintended Bias** |
> | ---------------------------------- | ---------------------- | ---------------------- |
> | Answer2 with Longer                | 100.00%                | 94.80%                 |
>
> **Fallacy-oversight Bias:**
>
> | **Principle-guided modifications** | **Bias Incorporation** | **No Unintended Bias** |
> | ---------------------------------- | ---------------------- | ---------------------- |
> | Answer1 with Fallacy               | 98.60%                 | 92.20%                 |
>
> **Sentiment Bias:**
>
> | **Principle-guided modifications** | **Bias Incorporation** | **No Unintended Bias** |
> | ---------------------------------- | ---------------------- | ---------------------- |
> | Answer1 with Cheerful              | 99.60%                 | 96.80%                 |
> | Answer1 with Sad                   | 99.00%                 | 93.80%                 |
> | Answer1 with Angry                 | 98.60%                 | 96.80%                 |
> | Answer1 with Fear                  | 99.20%                 | 93.00%                 |
> | Answer2 with Cheerful              | 98.40%                 | 97.40%                 |
> | Answer2 with Sad                   | 100.00%                | 95.60%                 |
> | Answer2 with Angry                 | 99.80%                 | 94.40%                 |
> | Answer2 with Fear                  | 100.00%                | 96.00%                 |
>
> **These results demonstrate the high reliability of our automated modification approach.** The complete human evaluation results and corresponding screenshots of our annotation platform have been added to the appendix of our PDF document, marked in blue text.
>
> Please refer to L972-995, Figure 14, and Table 10.

---

> ### Author Response · Authors · 2024-11-19
> **Response (2)**
>
> Q: Are there potential trade-offs between mitigating biases and maintaining the performance of LLM judges?
>
> A: Thank you for raising this important question. We believe that maintaining performance and mitigating biases in LLM judges are **not conflicting objectives but rather complementary goals**. Our reasoning is twofold:
>
> **A biased LLM judge cannot be considered a high-performing judge** - the very presence of bias undermines its fundamental role as an evaluator. Conversely, **an unbiased LLM judge naturally leads to more fair and accurate judgments**.
>
> In addition, based on our empirical findings, particularly for explicit biases, we can effectively **detect bias in the output text of Judge LLMs without compromising their performance**. This is demonstrated in Table 6 of our paper.
>
> Therefore, rather than viewing bias mitigation as a trade-off against performance, we see it as an essential component of improving the overall reliability and effectiveness of LLM judges.
>
> ------
>
> Dear Reviewer,
>
> We have addressed all concerns raised in the initial review comprehensively, including the incorporation of additional experiments and detailed explanations. Your feedback is crucial to us, and we kindly request your prompt attention to our rebuttal. **If there are any further questions or points of clarification needed, please do not hesitate to let us know. Your timely response would be greatly appreciated.**
>
> Once again, we appreciate your time and effort in reviewing our paper.

---

> > ### Comment · Reviewer_Tmem · 2024-11-20
> > **Thank you**
> >
> > Thanks for your detailed response. My concerns are mostly addressed and I will raise the score to reflect this.

---

> > > ### Author Response · Authors · 2024-11-20
> > > **Thanks a lot**
> > >
> > > We sincerely appreciate your time and dedication in reviewing our work, and are truly delighted by your strong endorsement of our research and rebuttal. Thanks a lot!

---

### Meta-Review · Area_Chair_GZRK · 2024-12-08

**Metareview:**

This paper introduces CALM, a novel framework to evaluate biases in LLMs used as judges for tasks like scoring and pairwise comparison. The authors identify and categorize 12 types of biases and assess them across six popular LLMs using diverse datasets and tailored metrics. While highlighting areas of fairness, the results reveal persistent biases in specific contexts.

This paper makes a timely and significant contribution to the NLP/LLM community by providing a comprehensive framework for identifying and analyzing biases in LLM judgments. While the metrics and result interpretations require refinement, the strengths of the paper, including its practical applicability and systematic exploration of biases, far outweigh the weaknesses. With minor revisions to address the issues around metrics and result clarity (which were addressed during the rebuttal phase), the paper would provide a valuable resource for researchers and practitioners. I recommend acceptance.

**Additional Comments On Reviewer Discussion:**

Several reviewers asked for additional experiments (e.g., additional metrics) and results analyses, which were perfectly addressed by the authors during the rebuttable phase.

---

### Decision · Program_Chairs · 2025-01-22

Accept (Poster)